# Counterfactual Graphical Models: Constraints and Inference

**Juan D. Correa** [1]   **Elias Bareinboim** [2]

## Abstract

Graphical models have been widely used as parsimonious encoders of the constraints underlying probability models. When organized in a structured way, these models can facilitate the derivation of non-trivial constraints, the inference of quantities of interest, and the optimization of their estimands. In particular, causal diagrams enable the efficient representation of the structural constraints of the underlying causal system. In this paper, we introduce an efficient graphical construction called *Ancestral Multi-world Networks* that is sound and complete for reading counterfactual independences from a causal diagram using d-separation. Moreover, we introduce the *counterfactual (ctf-) calculus*, which can be used to transform counterfactual quantities using three rules licensed by the constraints encoded in the diagram. This result generalizes Pearl's celebrated do-calculus from interventional to counterfactual reasoning.

## 1. Introduction

Counterfactuals form the basis of important notions across human cognition that require retrospective thinking, where one must compare what did happen in the real world versus what would have happened under some different hypothetical conditions. Given the impossibility of observing an alternative outcome once an action is taken, counterfactuals evoke "what if?" questions whose answers can only be approached by imagining hypothetical conditions contrary to this factual evidence. For instance, questions such as "what would be the death rates had the vaccination started two weeks earlier?" or "given that I arrived late, would I have been on time had I taken the subway instead of the taxi?" require us to carry out a mental experiment where

[1]Department of Computer Science, Universidad Autónoma de Manizales, Manizales, Colombia [2]Department of Commputer Science, Columbia University, New York, United States. Correspondence to: Juan D. Correa <jcorrea@autonoma.edu.co>.

*Proceedings of the 42nd International Conference on Machine Learning*, Vancouver, Canada. PMLR 267, 2025. Copyright 2025 by the author(s).

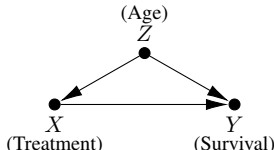

*Figure 1.* A causal diagram over three variables.

we recover some state of affairs, perform a change in the sequence of events, and let a hypothetical situation to play out. More generally, counterfactuals are an important component in the construction of explanations regarding why certain events occurred the way they did (Pearl, 2000; Pearl & Mackenzie, 2018; Bareinboim et al., 2020).

One fundamental topic of study in counterfactual reasoning is understanding the various quantities, the constraints on their relation, and the types of inferences allowed across various counterfactual worlds. Specifically, counterfactual quantities evoke hypothetical conditions that could contradict the factual evidence, underpinning different applications involving blame and responsibility, credit assignment, and more individualized types of decisions (Pearl, 2000). Examples of such quantities include the *effect of treatment on the treated* (Heckman, 1992; Pearl, 2000), *path-specific effects* (Pearl, 2001; Avin et al., 2005), and causal and spurious variations (Zhang & Bareinboim, 2018; Plečko & Bareinboim, 2024). There are also quantities such as the *probability of necessity* (PN), *probability of sufficiency* (PS), and the *probability of necessity and sufficiency* (PNS) that relate to fundamental aspects of how events are related and can explain the other. For example, consider the causal diagram in Figure 1 over the variables *age*, *treatment*, and *survival*. The counterfactual event $(Y_x = 1 \mid X = x')$ refers to the survival of a person ($Y = 1$) that gets a treatment $X = x$ when they would naturally decide not to get treated ($x'$). Such queries depict a quintessential counterfactual situation, since we aim to evaluate a world that contradicts the factual one in which the person was not treated.

In the first part of our paper, we revisit and generalize counterfactual constraints – exclusion and independence restrictions, and consistency (Pearl, 2000) – and show how they follow from the Structural Causal Model semantics. Specifically, we introduce a new graphical representation

that encodes independences between counterfactual random variables, which we call *Ancestral Multi-World Network* (AMWN). Based on this new data structure, we formally show that the d-separation criterion is complete for reading such constraints based on a causal graph and a set of counterfactual variables. Compared with the prior literature, the newly proposed method improves over the Twin Networks (Balke & Pearl, 1994), for which d-separation is not complete, and from Single World Intervention Graphs (Richardson & Robins, 2013), which consider a single intervention at a time. AMWN also differs from Multi-Networks and Counterfactual Graphs (Shpitser & Pearl, 2007), which were conjectured to be complete but require constructing a possibly exponential number of graphs to test separation among counterfactual variables rather than events.

In the second part of the paper, and building on the constraints and AMWN construction, we formulate a set of three rules for counterfactual inference called *Counterfactual Calculus* (ctf-calculus). Compared with the literature, our rules are more general than Pearl's celebrated do-calculus (Pearl, 1994; 1995) for interventional reasoning, since it allows for the transformation of counterfactual quantities to infer the implied equality constraints. Moreover, we show that the counterfactual calculus is complete for identifying counterfactuals from observational and interventional distributions. This set of rules also differs from the *Potential Outcome Calculus* (po-calculus) (Malinsky et al., 2019), which has been shown to hold if and only if the corresponding do-calculus rules hold. While po-calculus rules require counterfactual variables to follow certain patterns in terms of interventions and require pre-processing steps to be used for certain identification tasks, we propose rules supporting more general mixes of interventions, which, combined with probability axioms, are sufficient for deciding counterfactual identification.

More specifically, our contributions are as follows:

1. **Graphical criteria**: a sound, complete, and efficient procedure to test conditional independences among counterfactual variables using d-separation on a modified causal diagram.

2. **Inference rules**: a set of inference rules for counterfactual reasoning that are sound and complete for counterfactual identification from observational and experimental distributions.

Proofs can be found in the supplemental material.

**Definitions and Background.** We denote variables by capital letters, $X$, and values by small letters, $x$. Bold letters, $\mathbf{X}$ represent a set of variables and $\mathbf{x}$ a set of values. The domain of a variable $X$ is denoted by $\text{Val}(X)$. Two values

$\mathbf{x}$ and $\mathbf{z}$ are consistent if they share the common values for $\mathbf{X} \cap \mathbf{Z}$. We also denote by $\mathbf{x} \setminus \mathbf{Z}$ the value of $\mathbf{X} \setminus \mathbf{Z}$ consistent with $\mathbf{x}$ and by $\mathbf{x} \cap \mathbf{Z}$ the subset of $\mathbf{x}$ corresponding to variables in $\mathbf{Z}$. We assume the domain of every variable is finite.

We represent qualitative assumptions using causal graphs, denoted with a calligraphic letter, e.g., $\mathcal{G}$, etc. Given a graph $\mathcal{G}$, $\mathcal{G}_{\overline{\mathbf{W}}\underline{\mathbf{X}}}$ is the result of removing edges coming into variables in $\mathbf{W}$ and going out from variables in $\mathbf{X}$. $\mathcal{G}[\mathbf{W}]$ denotes a vertex-induced subgraph, which includes $\mathbf{W}$ and the edges among its elements. We use kinship notation for graphical relationships such as parents, children, descendants, and ancestors of a set of variables.

We base our analysis on the Structural Causal Model (SCM) paradigm (Pearl, 2000). An SCM $\mathcal{M}$ is a 4-tuple $\langle \mathbf{U}, \mathbf{V}, \mathcal{F}, P(\mathbf{u}) \rangle$, where $\mathbf{U}$ is a set of exogenous (latent) variables; $\mathbf{V}$ is a set of endogenous (observable) variables; $\mathcal{F}$ is a collection of functions such that each variable $V_i \in \mathbf{V}$ is determined by a function $f_i \in \mathcal{F}$. Each $f_i$ is a mapping from a set of exogenous variables $\mathbf{U}_i \subseteq \mathbf{U}$ and a set of endogenous variables $\mathbf{Pa}_i \subseteq \mathbf{V} \setminus \{V_i\}$ to the domain of $V_i$. Uncertainty is encoded through a probability distribution over the exogenous variables, $P(\mathbf{U})$.

An SCM $\mathcal{M}$ induces a *causal diagram* $\mathcal{G}$ where $\mathbf{V}$ is the set of vertices, there is a directed edge $(V_j \to V_i)$ for every $V_i \in \mathbf{V}$ and $V_j \in \mathbf{Pa}_i$, and a bidirected edge $(V_i \leftarrow\!-\!\rightarrow V_j)$ for every pair $V_i, V_j \in \mathbf{V}$ such that $U_i \cap U_j \neq \emptyset$ ($V_i$ and $V_j$ have a common exogenous parent) (Bareinboim et al., 2020). We assume that the underlying model is recursive. That is, there are no cyclic dependencies among the variables.

SCMs allow us to define counterfactual quantities with precision based on the *Pearl's Causal Hierarchy* (PCH) (Pearl & Mackenzie, 2018; Bareinboim et al., 2020). This hierarchy is divided into three layers (Figure 2): the first one ($\mathcal{L}_1$) captures the notion of "seeing," that is, observing a certain phenomenon or reality and possibly making inferences about it. The second ($\mathcal{L}_2$) allows one to represent the notion of "doing", that is, intervening (or deliberately acting) in the environment to bring about a certain state of affairs. Modifying an SCM gives natural valuations for quantities of this kind, as defined next.

**Definition 1.1** (Submodel). Let $\mathcal{M}$ be a causal model, $\mathbf{X}$ a set of variables in $\mathbf{V}$, and $\mathbf{x}$ a particular realization of $\mathbf{X}$. A submodel $\mathcal{M}_{\mathbf{x}}$ of $\mathcal{M}$ is the causal model

$$\mathcal{M}_{\mathbf{x}} = \langle \mathbf{U}, \mathbf{V}, \mathcal{F}_{\mathbf{x}}, P(\mathbf{U}) \rangle, \text{ where} \tag{1}$$

$$\mathcal{F}_{\mathbf{x}} = \{f_i : V_i \notin \mathbf{X}\} \cup \{\mathbf{X} \leftarrow \mathbf{x}\}. \tag{2}$$

That is, performing an external intervention (or action) is modeled through the replacement of the original (natural) mechanisms associated with some variables $\mathbf{X}$ with a constant $\mathbf{x}$, which is represented by the *do*-operator. The impact

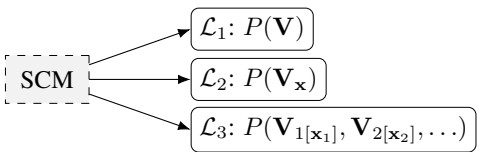

*Figure 2.* Every SCM induces different distributions in each layer of the PCH.

of the intervention on an outcome variable $Y$ is commonly called the *potential response*:

**Definition 1.2** (Potential Response). Let $\mathbf{X}$ and $\mathbf{Y}$ be two sets of variables in $\mathbf{V}$, and $\mathbf{u}$ be a unit. The potential response $\mathbf{Y_x}(\mathbf{u})$ is defined as the solution for $\mathbf{Y}$ of the set of equations $\mathcal{F_x}$ with respect to SCM $\mathcal{M}$ (for short, $\mathbf{Y}_{\mathcal{M_x}}(\mathbf{u})$). That is, $\mathbf{Y_x}(\mathbf{u}) = \mathbf{Y}_{\mathcal{M_x}}(\mathbf{u})$.

In other words, $\mathbf{Y}_{\mathcal{M_x}}(\mathbf{u})$ is obtained through the computation of $\mathbf{Y}(\mathbf{u})$ in the submodel $\mathcal{M_x}$. On the other hand, the meaning of every term in the counterfactual layer ($\mathcal{L}_3$) can be directly determined from a fully specified structural causal model, as described in the sequel:

**Definition 1.3** (Counterfactual Distribution Valuation). An SCM $\mathcal{M} = \langle \mathbf{U}, \mathbf{V}, \mathcal{F}, P(\mathbf{U}) \rangle$ induces a family of joint distributions over counterfactual events $\mathbf{Y_x}, \ldots, \mathbf{Z_w}$, for any $\mathbf{Y}, \mathbf{Z}, \ldots, \mathbf{X}, \mathbf{W} \subseteq \mathbf{V}$, $P^{\mathcal{M}}(\mathbf{y_x}, \ldots, \mathbf{z_w})$ is given by:

$$\sum_{\mathbf{u}} \mathbf{1}[\mathbf{Y_x}(\mathbf{u}) = \mathbf{y}, \ldots, \mathbf{Z_w}(\mathbf{u}) = \mathbf{z}] \ P(\mathbf{u}). \quad (3)$$

Let $\mathbf{W}_* = \{(W_1)_{\mathbf{T}_1}, (W_2)_{\mathbf{T}_2}, \ldots\}$ represent an arbitrary set of counterfactual variables such that $W_i \in \mathbf{V}$ and $\mathbf{T}_i \subseteq \mathbf{V}$ for $i = 1, \ldots, l$. We assume throughout this paper that all the distributions generated by the models are positive.

## 2. Counterfactual Constraints

We begin by stating three types of constraints that hold over counterfactuals random variables: **consistency** (Section 2.1), **exclusion** (Section 2.2), and **independence** (Section 2.3), which we detail in the following subsections.

### 2.1. Consistency Constraints

Consistency constraints relate to the interplay between observing a variable taking a particular value and the effect of an intervention that fixes this variable to the same value. To ground this idea, consider an SCM $\mathcal{M}$ over endogenous variables $\mathbf{V} = \{X, Y, Z\}$ and suppose we are interested in studying the joint counterfactual event $(Y_x = y, X = x)$. Following the proper semantics (Theorem 1.2), the value of variable $X$ is given by the solution of the system of equations $\mathcal{F}$ associated with $\mathcal{M}$, $X(\mathbf{u})$, for each unit $\mathbf{U} = \mathbf{u}$. Similarly, the value of $Y_x$ is given by the solution of the system $\mathcal{F_x}, Y_x(\mathbf{u})$, for the same unit. The event $X = x$ occurs

for $\mathbf{u}$ whenever the solution of $f_x$ is equal to $x$. While $f_x$ is fixed as a constant $x$ in $\mathcal{F_x}$ (as illustrated in Figure 3(a)), for any unit $\mathbf{U} = \mathbf{u}$ for which $X = x$, the result of these two systems of equations coincide.

Both models will match in the value of every observable, i.e., for $\mathbf{u}' = \{\mathbf{u} \mid X(\mathbf{u}) = x\}$,

$$X(\mathbf{u}') = X_x(\mathbf{u}') = x, Y(\mathbf{u}') = Y_x(\mathbf{u}'), \ Z(\mathbf{u}') = Z_x(\mathbf{u}'). \quad (4)$$

Moreover, the probability of the corresponding random variables follows from averaging $P(\mathbf{U})$ for those $\mathbf{u}$, and then:

$$P(Y_x = y, X = x)$$
$$= \sum_{\mathbf{u}} \mathbf{1}[Y_x(\mathbf{u}) = y, X(\mathbf{u}) = x] \ P(\mathbf{u}) \quad (5)$$
$$= \sum_{\mathbf{u}} \mathbf{1}[Y(\mathbf{u}) = y, X(\mathbf{u}) = x] \ P(\mathbf{u}) \quad (6)$$
$$= P(Y = y, X = x). \quad (7)$$

Again, this is so because $Y_x(\mathbf{u}) = Y(\mathbf{u})$ for those $\mathbf{u}$ for which $X(\mathbf{u}) = x$. More generally, when considering all the endogenous variables, we have:

$$P(Y_x, Z_x, X = x) = P(Y, Z, X = x), \quad (8)$$

In other words, once we restrict our attention to the set of units that generate $X = x$, then the variations of $Y_x$, and $Z_x$ are **consistent** with the variations of $Y$, and $Z$, respectively.

Intuitively, once $X$ takes the value $x$, naturally, other variables in the model behave the same as if $X$ had been fixed to $x$ by intervention, for instance, $Y_x = Y$.[1]

More broadly, consistency does not depend on the independence structure among the exogenous variables, $P(\mathbf{U})$, and follows from the relationships within the structural mechanisms $\mathcal{F}$.

The following characterizes this family of constraints across endogenous variables:

**Lemma 2.1** (Consistency). *Given SCM $\mathcal{M}$ and $X, Y \in \mathbf{V}$, $\mathbf{T}_*$ be any combination of counterfactuals, and let $x$ be a value in the domain of $X$. Then,*

$$P(Y_{\mathbf{T}_*}, X_{\mathbf{T}_*} = x) = P(Y_{\mathbf{T}_* x}, X_{\mathbf{T}_*} = x). \quad (9)$$

As suggested by the term $\mathbf{T}_*$ in Theorem 2.1, consistency between observations and interventions not only occurs for interventions that fix a variable to a constant value (e.g., $do\,(X = x)$) but is also true with interventions that set a variable to match another counterfactual variable, as discussed next.

---

[1] One way to interpret such a statement is through the independence of the mechanisms that give value to each of the endogenous variables in the system in conjunction with the locality of the intervention.

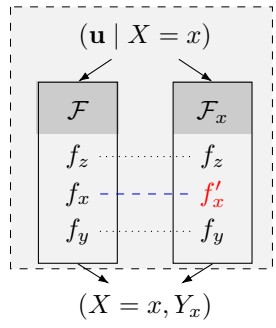
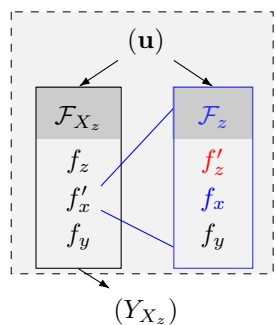

$(X = x, Y_x)$      $(Y_{X_z})$

(a) Mechanisms involved in generating the event $(Y_x, X = x)$.

(b) The variable $Y_{X_z}$ results from forcing $X$ to take the value $X_z(\mathbf{u})$ for every $\mathbf{u}$.

*Figure 3.* Representation of the mechanism involved in generating counterfactual events. The gray boxes represent data-generating mechanisms that transform a particular unit $\mathbf{U} = \mathbf{u}$ into a counterfactual event over the observable variables. Each rectangle is a copy of the mechanisms of the structural causal model. Depending on the counterfactual of interest, the mechanisms share some functions (e.g. $f_z$ and $f_y$ in (a)), redefine others ($f_x$ and $f'_x$ in (a)), or contain functions that require the evaluation of a separate set of mechanisms (e.g. $f'_x$ in (b)) to compute a nested counterfactual.

### 2.1.1. NESTED COUNTERFACTUALS

So far we have considered counterfactuals of the form $Y_x$, where the subscript $x$ indicates that an intervention $do\,(X = x)$ has been performed in the system. We turn our attention to interventions that could be expressed as $do\,(X = X_z)$, and represent settings where the variable $X$ is set to behave as another counterfactual variable, say $X_z$. This operation is illustrated in Figure 3(b). In other words, the value of $X_z$ is computed in a sub-model $F_z$ (figure's r.h.s.) and used to replace the natural mechanism $f_x$ (in the l.h.s.). A random variable $Y$ in such a system is represented with a counterfactual of the form $Y_{X_z}$, which is called a *nested counterfactual*.

The nesting means the target counterfactual internally refers to another nested world (possibly multiple times).

**Corollary 2.2** (Counterfactual Unnesting (CU)). *Let $Y, X \in \mathbf{V}$, $\mathbf{T}, \mathbf{Z} \subseteq \mathbf{V}$, and let $\mathbf{z}$ be a set of values for $\mathbf{Z}$. Then, the nested counterfactual $P(Y_{\mathbf{T}_* X_{\mathbf{z}}} = y)$ can be written with one less level of nesting as:*

$$P(Y_{\mathbf{T}_* X_{\mathbf{z}}} = y) = \sum_x P(Y_{\mathbf{T}_* x} = y, X_{\mathbf{z}} = x). \quad (10)$$

This statement follows from the law of total probability and consistency itself, i.e.:

$$P(Y_{\mathbf{T}_* X_{\mathbf{z}}} = y)$$

$$= \sum_x P(Y_{\mathbf{T}_* \boxed{X_{\mathbf{z}}}} = y, \boxed{X_{\mathbf{z}}} = x) \quad \text{(sum over } X_{\mathbf{z}}) \quad (11)$$

$$= \sum_x P(Y_{\mathbf{T}_* \boxed{x}} = y, X_{\mathbf{z}} = \boxed{x}) \quad \text{(consistency).} \quad (12)$$

These two steps allow us to reason about nested counterfactuals and transform them into expressions involving non-nested ones.

### 2.2. Exclusion Constraints

Although the semantics of counterfactuals allows one to consider a variable $Y_{\mathbf{t}}$ for arbitrary $Y \in \mathbf{V}$ and $\mathbf{T} \subseteq \mathbf{V}$, some counterfactual variables are not entirely free to vary depending on the topology and the sparsity of the causal system. For example, consider the simple chain graph in Figure 4(a) and the counterfactual variables $Y_z$ and $Y_{zx}$. To understand the relationship between these two variables, we write the corresponding sub-models $\mathcal{M}_z$ and $\mathcal{M}_{zx}$:

$$\mathcal{F}_z = \begin{cases} X_z \leftarrow f_X(U_x) \\ Z_z \leftarrow z \\ Y_z \leftarrow f_Y(z, U_y) \end{cases} \quad \mathcal{F}_{zx} = \begin{cases} X_{zx} \leftarrow x \\ Z_{zx} \leftarrow z \\ Y_{zx} \leftarrow f_Y(z, U_y), \end{cases}$$
$$(13)$$

and $P(\mathbf{U}) = P(U_x)P(U_z)P(U_y)$. Note that for each unit $\mathbf{U} = \mathbf{u}$, the variables $Y_z$ and $Y_{zx}$ are the same. Intuitively, once the value of $Z$ is fixed to $z$ by intervention, the only source of variation for the variable $Y$ in both $\mathcal{M}_z$ and $\mathcal{M}_{zx}$ comes from $U_y$, so intervening on $X$ is irrelevant. In some sense, the intervention on $X$ can be **excluded** without any changes in the value of $Y$, which gives the name exclusion restriction.

In graphical terms, an intervention on a variable $X$ could affect another variable $Y$ only if there exists a causal (directed) path from $X$ to $Y$ in $\mathcal{G}$.[2] Although in Figure 4(a) there is such path, the same is severed once $Z$ is intervened on. This observation can be stated more generally in the form of an operator used to exclude interventions from a given counterfactual variable as follows:

**Lemma 2.3** (Exclusion operator). *Let $Y_{\mathbf{x}}$ be a counterfactual variable, $\mathcal{G}$ a causal diagram, and*

$$Y_{\mathbf{z}} \text{ such that } \mathbf{Z} = \mathbf{X} \cap An(Y)_{\mathcal{G}_{\overline{\mathbf{X}}}} \text{ and } \mathbf{z} = \mathbf{x} \cap \mathbf{Z}. \quad (14)$$

*Then, $Y_{\mathbf{z}} = Y_{\mathbf{x}}$ holds for any model compatible with $\mathcal{G}$. Moreover, this transformation is denoted as $\|Y_{\mathbf{x}}\| := Y_{\mathbf{z}}$.*

Note that by keeping $\mathbf{X} \cap An(Y)_{\mathcal{G}_{\overline{\mathbf{X}}}}$ (Equation (14)), the exclusion operator removes from the counterfactual's antecedent (i.e., subscript) variables that are not ancestors of $Y$ (variables without causal paths to $Y$) as well as those ancestors that once $do\,(\mathbf{X})$ is performed are no longer ancestors of $Y$.

For a set $\mathbf{Y}_*$, define $\|\mathbf{Y}_*\| = \bigcup_{Y_{\mathbf{t}} \in \mathbf{Y}_*} \|Y_{\mathbf{t}}\|$. The result of applying the exclusion operator to $Y_{\mathbf{x}}$, $\|Y_{\mathbf{x}}\|$, is always equal to $Y_{\mathbf{x}}$ or an equivalent counterfactual variable with fewer variables in its antecedent.

---

[2]In terms of SCM, this means there is a sequence of functional substitutions such that $X$ may appear in the argument set of $Y$.

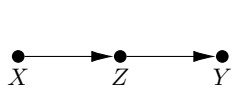

(a) Chain causal structure.

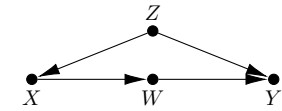

(b) Causal diagram over 4 variables.

*Figure 4.* Graphical structures used to illustrate exclusion and independence constraints.

One interesting feature of exclusion constraints is that they are derivable from the order relative to the mechanisms, $\mathcal{F}$, of the underlying SCM, $\mathcal{M}^*$. Other invariances from $\mathcal{M}^*$ come from sparsity in $P(\mathbf{U})$, as discussed in the sequel.

### 2.3. Independence Constraints and the Counterfactual d-separation Criterion

The ability to represent multiple worlds simultaneously is a fundamental aspect that sets apart the third layer of the PCH from the others. One could therefore consider a probability expression involving variables from multiple worlds, such as $Y_x$ and $Z_{x'}$ when $x \neq x'$.

At the structural level, multiple interventions entail different copies of the mechanisms $\mathcal{F}$ of the SCM, each for a different world (syntactically represented by a different subscript), but all sharing the same $P(\mathbf{U})$. As implied by Equation (3), a counterfactual distribution can be evaluated by passing the set of exogenous variables $\mathbf{U}$ through the different versions of those mechanisms, depending on which hypothetical world one aims to evaluate. This process can be mimicked and represented at the graphical level by a "meta" diagram incorporating different instances of the endogenous variables produced by the various mechanisms and connecting different worlds through the $\mathbf{U}$ variables. This idea allows the evaluation of separation statements among nodes representing counterfactual variables, which in turn imply conditional independences among the corresponding variables in the underlying distribution.

For concreteness, consider whether the causal graph in Figure 4(b) implies that $(Y_{xw}, W_{x'} \perp\!\!\!\perp X \mid Z_{x'})$. Figure 5(a) shows a natural generalization of the twin network to 3 worlds, a 3-plet network, for this graph and question. Note that the variables in the query involve three submodels: $\mathcal{M}, \mathcal{M}_{x'},$ and $\mathcal{M}_{xw}$, all depicted in the network sharing explicit unobservable variables.

While it seems that $X$ is d-connected to $Y_{xw}$ given $Z_{x'}$ in Figure 5(a), due to the active path $X \leftarrow Z \leftarrow U_z \rightarrow Z_{xw} \rightarrow Y_{xw}$, the exclusion operator reveals $Z_{x'} = \|Z_{x'}\| = Z$. This means that conditioning on $Z_{x'}$ is the same as conditioning on $Z$, and the separation holds.

In this sense, we should merge the nodes $Z, Z_{x'},$ and $Z_{xw}$ due to the deterministic relationship among them. It is also

convenient to ignore nodes of variables fixed by intervention and reduce every variable with the exclusion operator. This results in the 3-plet network shown in Figure 5(b). In this new graph, d-separation can be used to tell that $X$ and $Y_{xw} = Y_w$ are separated given $Z_{x'} = Z$.

More generally, we can construct twin networks, 3-plet networks, or k-plet networks depending on the number of interventions in the separation statement. Then, use the exclusion operator to merge nodes corresponding to variables that are deterministically the same. This method, however, includes many variables in the graph that we do not need to check. To improve efficiency, as we will show later, we discuss the concept of ancestors of a counterfactual.

**Definition 2.4** (Ancestors (of a counterfactual) (Correa et al., 2021)). Let $Y_{\mathbf{x}}$ be such that $Y \in \mathbf{V}, \mathbf{X} \subseteq \mathbf{V}$. Then, the set of (counterfactual) ancestors of $Y_{\mathbf{x}}$, denoted by $An(Y_{\mathbf{x}})$, consist of each $W_{\mathbf{z}}$ such that $W \in An(Y)_{\mathcal{G}_{\overline{\mathbf{X}}}} \setminus \mathbf{X}$ (which includes $Y$ itself), and $\mathbf{z} = \mathbf{x} \cap An(W)_{\mathcal{G}_{\overline{\mathbf{X}}}}$.

This extends the idea that, in graphical terms, a variable can only affect another if the former is an ancestor of the latter. When counterfactuals are involved, some of those ancestors become irrelevant (by virtue of the exclusion operator). For example, for the graph in Figure 4(a), $X$ is an ancestor of $Y$, but it is not a (counterfactual) ancestor of $Y_z$, because under $do\,(Z)$, $X$ cannot affect $Y_z$. Similarly, $Z$ is an ancestor of $Y$, but for $Y_x$ a counterfactual ancestor is $Z_x$, not $X$ or $Z$.

For a set of variables $\mathbf{W}_*$, we define $An(\mathbf{W}_*)$ as the union of the ancestors of each variable in the set. That is, $An(\mathbf{W}_*) = \bigcup_{W_{\mathbf{t}} \in \mathbf{W}_*} An(W_{\mathbf{t}})$. For example, in Figure 4(b), $An(Y_{xw}) = \{Y_w, Z\}, An(W_{x'}) = \{W_{x'}\}, An(X) = \{X, Z\}$.

We describe a graphical construction called the *Ancestral Multi-World Network* (AMWN), denoted $\mathcal{G}_A(\mathcal{G}, \mathbf{Y}_*)$. This data structure is a function of the original causal diagram $\mathcal{G}$ and the counterfactual variables $\mathbf{Y}_*$ in the separation statement to be evaluated. Algorithm 1 describes the procedure for creating an AMWN.

For concreteness, let us consider again the evaluation of the separation query $(Y_{xw}, W_{x'} \perp\!\!\!\perp X)$ using the causal diagram in Figure 4(b). In line 1, the procedure computes $An(\mathbf{Y}_*)$, which will be added as nodes in the AMWN.

The associated directed arrows witness the ancestrality of the variables involved. For instance, $Z$ is an ancestor (parent) of $Y_w$, hence $Z$ and the arrow $Z \rightarrow Y_w$ must be present in the graph. Note that, at this point, the resulting graph is a subgraph of Figure 5(b), but the rest of the graph is not relevant to evaluate the separation statement with d-separation. That is, $(X_{\mathbf{t}} \perp\!\!\!\perp Y_{\mathbf{r}} \mid \mathbf{Z}_*)$ can be judged using d-separation on top of $\mathcal{G}_A(X_{\mathbf{t}}, Y_{\mathbf{r}}, \mathbf{Z}_*)$.

The second part of Algorithm 1 explicitly adds latent vari-

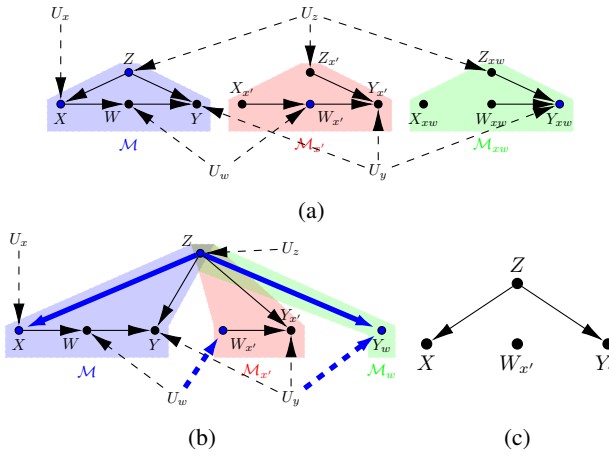

(a)

(b)                          (c)

*Figure 5.* Based on the causal diagram in Figure 4(b) and interventions $do(x, w)$, $do(x')$, and $do(\emptyset)$: 3-plet network (a), 3-plet network + exclusion. (c) AMWN with the ancestors of variables $Y_{xw}, W_{x'}, X$ and $Z$.

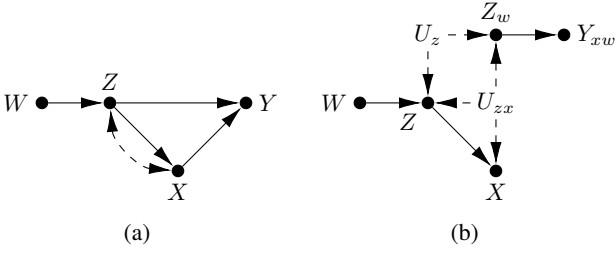

(a)                          (b)

*Figure 6.* To judge the separation statement $(Y_{xw} \perp\!\!\!\perp X \mid Z, W)$ in (a), we construct AMWN (b) and use d-separation.

ables $\mathbf{U}$ to facilitate reasoning about the relations among variables appearing more than once in the graph with different subscripts and variables originally connected by latent confounding.[3]

For instance, consider the causal diagram in Figure 6(a) and whether $(Y_{xw} \perp\!\!\!\perp X \mid \{Z, W\})$. The relevant set of ancestors is $An(Y_{xw}, X, Z, W) = \{Y_{xw}, Z_w, X, Z, W\}$. The corresponding AWMN is shown in Figure 6(b). The node $U_z$ has been added and connected to $Z$ and $Z_w$ (line 1), which come from the same original variable. There is also the node $U_{zx}$, that is connected to $Z, Z_w$ and $X$ due to the bidirected arrow $Z \leftarrow\!\!-\!\!\rightarrow X$ in $\mathcal{G}$ (line 1). By the d-separation criterion, the path $X \leftarrow\!\!-\!\!\rightarrow Z_w \rightarrow Y_{xw}$ is active given $\{Z, W\}$, which leads to the conclusion that $(Y_{xw} \not\perp\!\!\!\perp X \mid Z, W)$, as stated in the sequel.

**Theorem 2.5** (Independence Constraints — Counterfactual d-separation (soundness)). *Consider a causal diagram $\mathcal{G}$*

[3]The exogenous variables shared across worlds are precisely the anchors of invariance in these settings. They represent precisely the identity of the units submitted to these different counterfactual conditions.

**Algorithm 1** AMWN-CONSTRUCT$(\mathcal{G}, \mathbf{Y}_*)$

**Input**: Causal diagram $\mathcal{G}$ and a set of counterfactual variables $\mathbf{Y}_*$.
**Output**: $\mathcal{G}_A(\mathbf{Y}_*)$ the AMWN of $\mathcal{G}$ and $\mathbf{Y}_*$.
1: Initialize a network $\mathcal{G}'$ adding the variables in $An(\mathbf{Y}_*)$ together with the directed arrows witnessing the ancestrality.
2: **for each** variable $V \in \mathbf{V}$ appearing more than once in $\mathcal{G}'$ **do**
3:    Add a node $U_V$ and an edge $U_V \rightarrow V_{\mathbf{x}}$ for every instance of $V_{\mathbf{x}}$ of $V$.
4: **end for**
5: **for each** bidirected $V \leftarrow\!\!-\!\!\rightarrow W$ where $V$ and $W$ are in $\mathcal{G}'$ **do**
6:    Add a node $U_{VW}$ and edges from it to $V_{\mathbf{x}}$ and $W_{\mathbf{x}}$, for every instance of $V_{\mathbf{x}}$ of $V$ or $W_{\mathbf{x}}$ of $W$ in $\mathcal{G}'$.
7: **end for**
8: **return** $\mathcal{G}'$.

and a collection of counterfactual distributions, $\mathbf{P}_{***}$, induced by the SCM associated with $\mathcal{G}$. For counterfactual variables $X_{\mathbf{t}}, Y_{\mathbf{r}}, \mathbf{Z}_*$,

$$(\|X_{\mathbf{t}}\| \perp\!\!\!\perp \|Y_{\mathbf{r}}\| \mid \|\mathbf{Z}_*\|)_{\mathcal{G}_A} \rightarrow (\|X_{\mathbf{t}}\| \perp\!\!\!\perp \|Y_{\mathbf{r}}\| \mid \|\mathbf{Z}_*\|)_{\mathbf{P}_{***}}.$$

*Moreover, if an independence relationship cannot be inferred with the criterion, there exist at least two models inducing the graphical model where the independence does not hold.*

In other words, the d-separation criterion is sound for the counterfactual variables in the AMWN, that is, if $\|X_{\mathbf{t}}\|$ and $\|Y_{\mathbf{r}}\|$ are d-separated given $\|\mathbf{Z}_*\|$ in the diagram $\mathcal{G}_A(X_{\mathbf{t}}, Y_{\mathbf{r}}, \mathbf{Z}_*)$, then $X_{\mathbf{t}}$ and $Y_{\mathbf{r}}$ are independent given $\mathbf{Z}_*$ in every distribution $\mathbf{P}_{***}$ compatible with the causal diagram $\mathcal{G}$. It is also complete because if d-separation does not hold in the AMWN, then independence cannot be guaranteed.

This result allows us to use the construction of an AMWN of $\mathcal{G}$ to test whether a pair of counterfactual variables is independent in the probability distributions generated by the model compatible with $\mathcal{G}$.

Now, we examine the time complexity of constructing an AMWN. Let $z$ be the number of different interventions in the separation query, and $n$, $m$ be the number of nodes and edges, respectively. In line 1, the set of counterfactual ancestors can be computed in time linear to the size of the graph, for each intervention appearing in $\mathbf{Y}_*$; hence the step takes time $O(z(n + m))$. Due to line 1, no more than $n$ latent nodes and $zn$ edges are added. Line 1 adds $m$ latent nodes and $2zm$ edges at most. Overall, the construction takes $O(z(n + m))$, which is polynomial in the size of $\mathcal{G}$ and $\mathbf{Y}_*$.

The resulting graph $\mathcal{G}_A$ has $O(z(n + m))$ nodes and edges, hence running d-separation on top of it takes $O(z(n + m))$ time (van der Zander et al., 2014). Compared with the classical d-separation criterion, the time required to use AMWN increases by a factor of $z$, the number of different worlds involved in the query. Table 1 summarizes the methods

discussed, in terms of whether they allow for checking any separation constraints (among any counterfactual in the considered worlds), if d-separation is complete for them, and the time complexity of the construction of the graph and checking the constraint.[4] Overall, the method based on AMWN is more general and efficient than previous algorithms in the literature.

## 3. The Counterfactual Calculus

Building on our understanding of the constraints discussed earlier, this section introduces the counterfactual calculus and how it can be used for counterfactual inference based on the assumptions encoded in a causal diagram.

In the spirit of Pearl's celebrated interventional calculus (do-calculus), its counterfactual counterpart allows one to transform expressions in the form $P(\mathbf{y}_* \mid \mathbf{x}_*)$ to other counterfactual quantities, including in observational ($P(\mathbf{y} \mid \mathbf{x})$) and experimental ($P(\mathbf{y} \mid do(\mathbf{x}))$) forms, as licensed by the constraints encoded in the causal diagram. The counterfactual calculus consists of three inference rules based on the three types of constraints discussed earlier.

**Theorem 3.1** (Counterfactual Calculus (ctf-calculus)). *Let $\mathcal{G}$ be a causal diagram, then for $\mathbf{Y}, \mathbf{X}, \mathbf{Z}, \mathbf{W}, \mathbf{T}, \mathbf{R} \subseteq \mathbf{V}$, the following rules hold for the probability distributions generated by any model compatible with $\mathcal{G}$:*

**Rule 1** *(Consistency rule — Obs./intervention exchange)*

$$P(\mathbf{y}_{\mathbf{T}_*\mathbf{x}}, \mathbf{x}_{\mathbf{T}_*}, \mathbf{w}_*) = P(\mathbf{y}_{\mathbf{T}_*}, \mathbf{x}_{\mathbf{T}_*}, \mathbf{w}_*) \qquad (15)$$

**Rule 2** *(Independence Rule — Adding/removing counterfactual observations)*

$$P(\mathbf{y}_{\mathbf{r}} \mid \mathbf{x}_{\mathbf{t}}, \mathbf{w}_*) = P(\mathbf{y}_{\mathbf{r}} \mid \mathbf{w}_*)$$
$$\text{if } (\mathbf{Y}_{\mathbf{r}} \perp\!\!\!\perp \mathbf{X}_{\mathbf{t}} \mid \mathbf{W}_*) \text{ in } \mathcal{G}_A, \quad (16)$$

**Rule 3** *(Exclusion Rule — Adding/removing interventions)*

$$P(\mathbf{y}_{\mathbf{xz}}, \mathbf{w}_*) = P(\mathbf{y}_{\mathbf{z}}, \mathbf{w}_*)$$
$$\text{if } \mathbf{X} \cap An(\mathbf{Y}) = \emptyset \text{ in } \mathcal{G}_{\overline{\mathbf{Z}}}, \quad (17)$$

*where $\mathcal{G}_A$ is the counterfactual ancestral graph $\mathcal{G}_A(\mathcal{G}, \mathbf{Y}_{\mathbf{r}} \cup \mathbf{X}_{\mathbf{t}} \cup \mathbf{W}_*)$.*

The first rule of the calculus, consistency, was discussed in Section 2.1. One distinct feature of this rule is that it does not depend on the graphical structure and allows for adding or removing interventions whenever a specific observational context and the antecedent of the counterfactual (subscript)

| Method | Any sep. | Complete | Time Complexity[5] |
|---|---|---|---|
| Twin Network | Yes | No | $O(n+m)$ |
| SWIG | No | Yes | $O(n+m)$ |
| Multi-Networks | Yes | Conject. | $O(d^n(n+m))$ |
| **k-plet Network** | Yes | Yes | $O(zn(n+m))$ |
| **AMWN** | Yes | Yes | $O(z(n+m))$ |

*Table 1.* Comparison of counterfactual independence graphical constructions. For each method, we look at whether it supports any separation query, whether it is complete, and its time complexity.

match. As mentioned earlier, consistency is essentially the probabilistic instantiation of the invariances that follow from the modularity and stability of the causal mechanisms of the underlying system.

The second rule, *independence* (Section 2.3), corresponds to a generalized version of d-separation for counterfactual events. Syntactically, it permits the addition/removal of counterfactual evidence in a probability distribution.

The third rule, *exclusion* (Section 2.2), follows from the idea that interventions on variables without a causal path to the observed variable do not affect this variable and, therefore, can be dismissed.[6]

For concreteness, we illustrate next the use of the ctf-calculus rules for counterfactual identification tasks through a few examples.

*Example* 1 (ETT in the Backdoor diagram). Consider the causal diagram in Figure 1 and the observational distribution as input, and the counterfactual distribution $P(y_x \mid x')$ as the query. Using the ctf-calculus, we can then write:

$$P(y_x \mid x')$$
$$= \sum_z P(y_x \mid z, x')P(z \mid x')$$
$$\text{(Conditioning on } Z) \qquad (18)$$
$$= \sum_z P(y_x \mid z_x, x')P(z \mid x')$$
$$\text{(R3: } \{X\} \cap An(Z) = \emptyset) \qquad (19)$$
$$= \sum_z P(y_{xz} \mid z_x, x')P(z \mid x')$$
$$\text{(R1: } (Z_x = z \Rightarrow Y_x = Y_{xz})) \qquad (20)$$
$$= \sum_z P(y_{xz} \mid z, x')P(z \mid x')$$
$$\text{(R3: } \{X\} \cap An(Z) = \emptyset) \qquad (21)$$
$$= \sum_z P(y_{xz} \mid z, x)P(z \mid x')$$
$$\text{(R2: } (X \perp\!\!\!\perp Y_{xz} \mid Z) \text{ in } \mathcal{G}_A \text{ (Figure 7(a)))}$$
$$\qquad (22)$$
$$= \sum_z P(y \mid z, x)P(z \mid x')$$

---

[4] Further details on this comparison are given in Appendix C.1 in (Correa & Bareinboim, 2024).

[5] $n, m, z$, and $d$ refer to the number of nodes, edges, (different) interventions, and maximum cardinality of any observable variable in $\mathcal{G}$, respectively.

[6] We provide a comparison of ctf-calculus and do-calculus in Appendix C.2 in (Correa & Bareinboim, 2024).

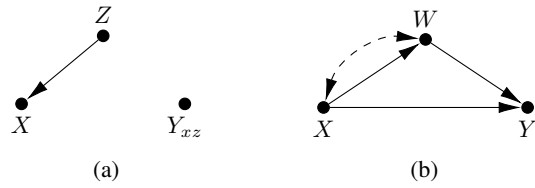

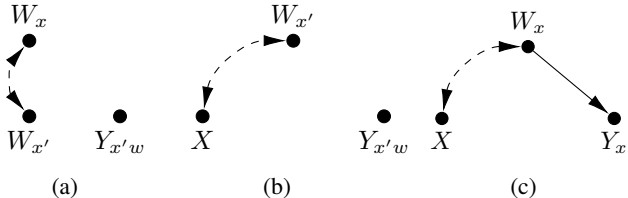

*Figure 7.* (a) An AMWN based on Figure 1 and variable set $\mathbf{Y}_* = \{Y_{xz}, X, Z\}$ used in the derivation of Example 1. (b) causal diagram used in Example 2.

*Figure 8.* Causal diagrams used in derivation in Example 2.

$$(\text{R1: } (Z = z, X = x \Rightarrow Y_{xz} = Y)) \quad (23)$$

The effect of the treatment on the treated is then identifiable from $P(x, z, y)$ and $\mathcal{G}$. ∎

*Example* 2 (Natural Direct and Indirect Effects). Consider the causal diagram in Figure 7(b) and suppose we wish to evaluate the natural direct (NDE) to understand how exercising ($X$) affects cardiovascular disease ($Y$) by means other than affecting the cholesterol level ($W$), that acts as a mediator of this relationship. The NDE can be written in counterfactual language as

$$NDE_{x,x'}(y) = P(y_{x',W_x}) - P(y_x). \quad (24)$$

The derivation of the first term of the NDE expression goes as follows:

$$P(y_{x'W_x})$$
$$= \sum_w P(y_{x'w}, w_x)$$
$$\quad (\text{CU, cor. 2.2: sum over } W_x + \text{consistency}) \quad (25)$$
$$= \sum_w P(y_{x'w} \mid w_x)P(w_x)$$
$$\quad (\text{Chain rule}) \quad (26)$$
$$= \sum_w P(y_{x'w} \mid w_{x'})P(w_x)$$
$$\quad (\text{R2: } (Y_{x'w} \perp\!\!\!\perp W_x, W_{x'}) \text{ in } \mathcal{G}_A \text{ (Figure 8(a))}) \quad (27)$$
$$= \sum_w P(y_{x'w} \mid w_{x'}, x')P(w_x)$$
$$\quad (\text{R2: } (Y_{x'w} \perp\!\!\!\perp X \mid W_{x'}) \text{ in } \mathcal{G}_A \text{ (Figure 8(b))}) \quad (28)$$
$$= \sum_w P(y_{x'w} \mid w, x')P(w_x)$$
$$\quad (\text{R1: } (X = x' \Rightarrow W_{x'} = W)) \quad (29)$$
$$= \sum_w P(y \mid w, x')P(w_x)$$
$$\quad (\text{R1: } (W = w, X = x' \Rightarrow Y_{x'w} = Y)). \quad (30)$$

Here, $P(w_x)$ cannot be further reduced to an expression in terms of observational distributions.

For the baseline $P(y_x)$, the derivation goes as follows:

$$P(y_x)$$
$$= \sum_w P(y_x \mid w_x)P(w_x)$$

$$(\text{Condition on } W_x) \quad (31)$$
$$= \sum_w P(y_x \mid w_x, x)P(w_x)$$
$$\quad (\text{R2: } (Y_x \perp\!\!\!\perp X \mid W_x) \text{ in } \mathcal{G}_A \text{ (Figure 8(c))}) \quad (32)$$
$$= \sum_w P(y \mid w, x)P(w_x)$$
$$\quad (\text{R1: } (X = x \Rightarrow W_x = W, Y_x = Y)). \quad (33)$$

Finally, we get

$$\text{NDE}_{x,x'}(y) = \sum_w \left(P(y|w,x') - P(y|w,x)\right)P(w_x). \quad (34)$$

by putting Equations (30) and (33) together. ∎

The calculus guarantees the correctness of the reduction whenever such a derivation from a counterfactual query to the probabilities over the observed distributions is available.

**Theorem 3.2** (Soundness and Completeness for Counterfactual Identifiability). *A counterfactual quantity $\mathcal{Q} = P(\mathbf{y}_* \mid \mathbf{x}_*)$ is identifiable from a given combination of observational and experimental distributions and a causal diagram $\mathcal{G}$ if and only if there exists a sequence of applications of the rules of ctf-calculus and the probability axioms that reduces $\mathcal{Q}$ into a function of the available distributions.*

In other words, any derivation following the ctf-calculus is correct (soundness) and, if any counterfactual is identifiable from certain observational ($\mathcal{L}_1$) and interventional ($\mathcal{L}_2$) distributions, there must exist a sequence of applications of the ctf-calculus that witnesses the mapping of the available distributions and the target effect (completeness).

Since any causal effect can be written in counterfactual terms, it is only natural that ctf-calculus subsumes do-calculus, as follows.

**Lemma 3.3** (ctf-calculus — do-calculus reduction). *ctf-calculus subsumes do-calculus.*

More details on the relationship between do-calculus and ctf-calculus can be found in Appendix C.2 in (Correa & Bareinboim, 2024).

## 4. Conclusions

In this paper, we first established consistency (Theorem 2.1), exclusion (Theorem 2.3), and independence constraints (Theorem 2.5) following from the SCM semantics. We showed that d-separation is complete for obtaining independence constraints from the causal diagram using an efficient graphical construction called Ancestral Multi-World Network (Algorithm 1). This constitutes the first efficient procedure for reading counterfactual independence. We then introduced a set of rules called *counterfactual calculus* (Theorem 3.1), which can be used to transform target counterfactual quantities based on the constraints encoded in the diagram. Finally, we showed that counterfactual calculus is sound and complete for identifying counterfactuals from an arbitrary combination of observational and experimental distributions (Theorem 3.2). We hope the results in this paper can further our understanding and expand the toolbox for performing causal reasoning, closing a journey that started with Pearl's fundamental results on d-separation for observational distributions (circa 1986) and the do-calculus for interventional reasoning (1995). We now have more general machinery that allows for reasoning across the three layers of the causal hierarchy, including the very top: counterfactual relations.

## Acknowledgements

This research is supported in part by the NSF, ONR, AFOSR, DoE, Amazon, JP Morgan, and The Alfred P. Sloan Foundation.

## Impact Statement

This paper presents results that advance the field of Machine Learning, particularly in the area of Causal Inference. There are many potential societal consequences of our work, none of which we feel must be specifically highlighted here.

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
