# OpenReview forum: "Counterfactual Graphical Models: Constraints and Inference"
_ICML.cc/2025/Conference — ICML 2025 spotlightposter_

### Official Review · Reviewer_JAeX · 2025-03-13

**Overall Recommendation:** 3

**Summary:**

The paper presents a novel framework for counterfactual reasoning using graphical models. The paper introduces two key contributions: Ancestral Multi-World Networks (AMWN) – a new graphical representation for counterfactuals, and Counterfactual Calculus (ctf-calculus) – a set of rules for transforming counterfactual expressions. The work extends Pearl’s do-calculus, allowing for more efficient and general counterfactual reasoning.
AMWN is a graphical model that encodes counterfactual independence relationships. It builds upon Structural Causal Models (SCMs) and replaces traditional Twin Networks, which are computationally expensive. The paper provides an algorithm to construct AMWNs, ensuring they are both sound (correct) and complete (able to capture all relevant relationships). The model enables efficient testing of counterfactual independence using d-separation.
Counterfactual Calculus (ctf-calculus) is a set of three transformation rules that generalize do-calculus to counterfactual settings: First, the consistency rule relates observations and interventions. Second, the independence rule uses d-separation to infer counterfactual independence. Third, the exclusion rule eliminates interventions that do not affect a variable.
These rules allow counterfactual expressions (e.g., $P(Y_x | X=x’)$ ) to be rewritten in terms of observable or interventional probabilities.

The approach is computationally efficient, reducing the complexity of counterfactual queries compared to existing methods.

**Claims And Evidence:**

The paper presents several claims that can be summarized in two areas. First, it claims AMWN is a sound and complete method for counterfactual independence reasoning and improves on existing graphical models (e.g., Twin Networks, Single World Intervention Graphs).
These assertions are supported by consistency and constraint definitions, nested (and unnesting) counterfactuals, exclusion operators, and a hierarchy of counterfactual relations. These are used for the key theorem on counterfactual d-separation (independence)
Second, the paper presents the CTF as a generalization of Pearl’s do-calculus (Theorem 3.1), resulting from counterfactual d-separation.

**Essential References Not Discussed:**

I would suggest to discuss the counterfactual d-separation concept in the context of other graphical models criteria of independence. See for instance Ma et al, (2022), Vo et al. (2023) , etc.

Ma, Jing, et al. "Clear: Generative counterfactual explanations on graphs." Advances in neural information processing systems 35 (2022): 25895-25907.
Vo, Vy, et al. "Feature-based learning for diverse and privacy-preserving counterfactual explanations." Proceedings of the 29th ACM SIGKDD Conference on Knowledge Discovery and Data Mining. 2023.

**Experimental Designs Or Analyses:**

The paper does not include empirical experiments or simulations

**Methods And Evaluation Criteria:**

The proposed methods are theoretical. They introduce AMWN as a new representation of counterfactual graphical models. The paper defines ctf-calculus as a set of inference rules using d-separation in the AMWN framework to determine counterfactual independence.
Since the work is theoretical, the evaluation criteria consist of the proofs. This includes computational complexity analysis compared with prior methods in terms of efficiency (Twin Networks, SWIG, and Multi-Networks).

**Other Comments Or Suggestions:**

Just a suggestion to rename the heading Any sep. in Table 1 for something more informative. Alternatively, it could be added to the caption.

**Other Strengths And Weaknesses:**

Two observations on this work. First, the paper provides a general framework for counterfactual reasoning, unifying different causal inference tools. Second, the method is computational efficiency as AMWN provides a polynomial-time method for counterfactual independence testing.

**Questions For Authors:**

How does your proposed idea compare to the works listed above?

**Relation To Broader Scientific Literature:**

The paper extends prior work on Pearl’s do-calculus and Structural Causal Models (SCMs). It improves on existing methods such as Balke & Pearl, 1994); AMWN overcomes the incompleteness of d-separation in Twin Networks; Richardson & Robins (2013), for which AMWN handles multiple interventions simultaneously; Shpitser & Pearl (2007); for which AMWN avoids the exponential graph explosion problem.
The literature review is strong, but an empirical comparison with recent causal inference models (e.g., deep learning-based causal models) could be useful.

**** POST REBUTTAL

thank you to the authors for their answer. Their comments can be helpful in improving the paper. I will maintain my positive score for this paper.

**Theoretical Claims:**

The key claims are summarized in three theorems, among other lemmas. Theorem 2.5 (Counterfactual d-Separation) proves that d-separation in AMWN is both sound and complete. Theorem 3.1 (Counterfactual Calculus Rules) formally establishes the transformation rules for counterfactual expressions. Theorem 3.2 (Soundness and Completeness of ctf-calculus for Counterfactual Identifiability) shows that ctf-calculus can fully characterize counterfactual identification.

---

> ### Author Rebuttal · Authors · 2025-04-01
>
> Thank you for reading our work, providing feedback, and asking questions.
>
> We refer next to the research mentioned in the review. Also, thank you for sharing the references. We describe the work as we understand it, but we would be happy to hear more about it from the reviewer. Regarding Ma et al (2022), the paper makes assumptions on the latent variables’ prior and the availability of an auxiliary variable. Based on these, they aim to identify the SCM that generates the data. They integrate the information on the SCM into the optimization process of a generative model to produce explanations. Compared to our approach, there are similarities in modeling the data-generating process as an SCM and then inferring counterfactual quantities based on the assumptions made over the SCM. On the other hand, in our framework, we make no assumptions about the distribution of the latent variables or the functional form of the mechanism of the SCM but assume a known causal graph. This setting is called non-parametric in the literature, which motivated Pearl’s Biometrika’s 1995 paper, where he introduced the do-calculus. Moreover, we do not attempt to identify the SCM as a whole but to identify particular (counterfactual) queries of interest in the form of independence constraints or probabilities. These are indeed the two main contributions of our paper, namely, graphical criteria and calculus for counterfactual identification.
>
> Vo et al (2023) seems to focus on producing examples that counter the original outcome produced by a model (e.g. classifier). Moreover, it seems the validity of the counterfactual is measured in terms of whether the outcome can be overturned by the generated example. We believe our work is not directly comparable here since it seems this work does not address the causal structure of the underlying data-generating process. Having said that, we would be happy to understand the reviewer's suggestions, in case it implies some subtle connection we may have missed based on our cursory reading.

---

> > ### Comment · Reviewer_JAeX · 2025-04-09
> >
> > Thank you for the answer. I will maintain my score.

---

### Official Review · Reviewer_ajna · 2025-03-13

**Overall Recommendation:** 4

**Summary:**

The paper studies the identification of counterfactual queries. It studies the constraints induced by the casual graph: consistency, exclusion, and independence. The paper proposes a sound and complete method for testing independencies among counterfactual variables based on constructing a simplified graph (AMWN) and then testing d-separations on the graph. These constraints then lead to a set of sound and complete rules, called counterfactual calculus, for identifying counterfactual queries.

**Claims And Evidence:**

Yes, proofs for all theorems were included in the supplementary.

**Essential References Not Discussed:**

I'm not aware of any missing references.

**Experimental Designs Or Analyses:**

N/A

**Methods And Evaluation Criteria:**

Examples were provided in the paper to illustrate how the counterfactual calculus can be applied to identify counterfactual queries.

**Other Comments Or Suggestions:**

- Figure 3 is a bit confusing to me since the meaning of the two columns and edges is missing.
- While this was shown in the Appendix, I think it is worth mentioning in the main paper that counterfactual calculus can remove interventions (do operations) whenever they can be removed using do-calculus. Otherwise, it's unclear how it is complete for identification given the observational distributions.

Typos:
- Page 2: "We base our analysis on the Structural Causal Model (SCM) paradigm *(?)Ch. 7]pearl:2k.*"
- Page 6, "Algorithm 1: For each *edge* $V \in \textbf{V}$"    ->   variable?
- Supplementary Definition D.2:  "conditions ??",   Section E Q1: "(see (?)Sec. 1.3]bar:etal2020 for details"

**Other Strengths And Weaknesses:**

- The paper is clearly structured and well-written.
- The supplementary contains a huge discussion on the connection to other related frameworks.
- The paper also included the examples (end of Sec. 3), which makes it easier for me to see how the rules can be applied.

**Questions For Authors:**

Could you please provide some intuitions for the (counterfactual) ancestors in Definition 2.4? Maybe include it after the definition.

**Relation To Broader Scientific Literature:**

The paper proposes a sound and complete method for testing independencies for general counterfactual variables, which improves upon the previous methods including [Balke & Pearl, 1994] (not complete), [Shpitser & Pearl 2007] (not efficient), and [Richardson & Robins, 2013] (restricted to a single world).

Moreover, the paper proposes sound and complete rules for identifying counterfactual queries, which complement the previous algorithmic method in [Correa et al., 2021a]. As mentioned in Appendix E, this is an important contribution since the rules (counterfactual calculus) bring the potential to solve problems under more general setups.

**Theoretical Claims:**

I reviewed the proofs in Appendix B.1, B.3, B.5, and C.2 in detail and skimmed through others. They look correct and rigorous to me.

---

> ### Author Rebuttal · Authors · 2025-04-01
>
> Thank you for reviewing our work, providing feedback, and giving suggestions.
>
> To answer your question about the intuition for counterfactual ancestors (Definition 2.4), they are the counterfactual variables that are causally relevant to the variable in question. This extends the idea that, in graphical terms, a variable can only affect another if the former is an ancestor of the latter. When counterfactuals are involved, some of those ancestors become irrelevant (by virtue of the exclusion operator). For example, in a simple chain graph such as $X \to Z \to Y$, $X$ is an ancestor of $Y$, but it is not a (counterfactual) ancestor of $Y_z$, because under $do(Z)$, $X$ cannot affect $Y_z$. Similarly, $Z$ is an ancestor of $Y$, but for $Y_x$ the counterfactual ancestor is $Z_x$, not $X$ or $Z$. As suggested, we will include a brief discussion of this after the definition; your suggestion is appreciated.
>
> As you point out, we did not mention in the main paper that ctf-calculus subsumes do-calculus. It is suitable for any task where the latter is complete, such as identification from observational distributions. We will add Lemma C.1 (in the supplemental material) “ctf-calculus subsumes do-calculus” and a brief discussion to the main text to make this fact clear to the reader.
>
> We see your point about Figure 3, and that the meaning of the columns and edges can be confusing in the middle of so many models where elements have a well-defined meaning. In this figure, the gray boxes represent data-generating mechanisms that transform a specific unit $U=u$ into a counterfactual event over the observable variables. Each rectangle is a copy of the mechanisms of the structural causal model. Depending on the counterfactual of interest, the mechanisms share some functions (e.g., $f_z$ and $f_y$ in (a)), redefine others ($f_x$ and $f_x'$ in (a)), or contain functions that require the evaluation of a separate set of mechanisms (e.g. $f_x'$ in (b)) to compute a nested counterfactual. Following your suggestion, we will provide a better description of the graphical elements of the figure in the manuscript, thank you.

---

### Official Review · Reviewer_RfwY · 2025-03-13

**Overall Recommendation:** 5

**Summary:**

The paper is focused on the graphical modelling and the (symbolic) calculus of counterfactual inferences within the framework of Pearlian structural causal models. There are two major contributions: (i) a novel graphical representation called Ancestral Multi-World Networks (AMWN), which efficiently encodes counterfactual independencies implied by causal diagrams (d-separation is complete wrt AMWNs); (ii)
a new set of inference rules called "counterfactual calculus" that extend Pearl's classical do calculus to counterfactuals (also sound and complete for those queries).

**Claims And Evidence:**

All the claims about the construction and soundness of the new graphical structure and the corresponding calculus are formally proved.

**Essential References Not Discussed:**

I think all the relevant references are properly cited.

**Ethical Review Concerns:**

-

**Experimental Designs Or Analyses:**

This is a theoretical paper with no experiments.

**Methods And Evaluation Criteria:**

This is a theoretical paper with no experiments.

**Other Comments Or Suggestions:**

The sentence about "transforming nested ctf into non-nested one" might be misleading, as we have a sum in the transformation.
There are a few typos in the references (ex. "(?)Ch. 7]pearl:2k"). Moreover, in section 2.3., the test query is conditioned on Z intervened by setting X=x’ and not Z’ intervened setting X=x.
Some of the material in S2 is not entirely novel. The authors should better make this explicit in their revised version.
Has the lack of completeness of twin nets been explicitly stated in the original paper of Balke & Pearl?

As I can understand, the idea of computing a CTF query in the twin network after the surgery might therefore lead to wrong conclusions. If so, it would be nice to emphasize this point in the paper.

**Other Strengths And Weaknesses:**

This is a significant and influential paper for counterfactual inference. The work fills a gap in the earlier literature, and I might imagine lots of applications based on the calculus presented here. Of course, some results are pretty technical, but this is expected and the authors did an excellent job in giving additional information and insights in the supplementary material.

**Questions For Authors:**

-

**Relation To Broader Scientific Literature:**

The relation with the broader scientific literature is very clear.

**Theoretical Claims:**

I checked the proofs of the main results but not those of the preliminary lemmas. I believe the results are correct.

---

> ### Author Rebuttal · Authors · 2025-04-01
>
> Thank you for reading our work, pointing out typos, and providing suggestions, which we will incorporate into the manuscript.
> In particular, we will clarify in the paper that the unnesting corresponds to a transformation that starts with a nested counterfactual and ends with an expression involving a counterfactual with one less level of nesting.
>
> As far as we can tell, the original paper describing the Twin Network method by Balke & Pearl does not claim the method’s completeness for using d-separation to assess counterfactual independencies. However, in Shpitser & Pearl (2007), the authors discuss the incompleteness of the method to motivate multi-networks.
>
> As you point out, “the idea of computing a CTF query in the twin network after the surgery might therefore lead to wrong conclusions”. For instance, in the example in Sec. 2.3, related to Figure 5 (a,b), some distinct nodes in the Twin Network may refer to counterfactual variables that are deterministically the same. In the example, conditioning on $Z_{x}'$ is the same as conditioning on $Z$. Because the Twin Network does not capture this constraint graphically, d-separation does not capture such independencies. We have also provided further discussion on this in section C of the supplemental material but will clarify this in the main text as well. Thank you!

---

### Official Review · Reviewer_GANy · 2025-03-14

**Overall Recommendation:** 4

**Summary:**

The paper introduces an efficient graphical construction called Ancestral Multi-world Network, which is sound and complete for interpreting counterfactual independencies from a causal diagram through d-separation. Furthermore, the authors propose the counterfactual (ctf-) calculus, which provides three transformation rules for deriving counterfactual quantities based on the constraints encoded within the diagram.

**Claims And Evidence:**

Yes, the claims presented in the paper are supported by clear and convincing evidence.

**Essential References Not Discussed:**

Although the paper mentions the k-plet Network, it appears that no relevant references are provided.

**Experimental Designs Or Analyses:**

There is no experimental designs or analyses.

**Methods And Evaluation Criteria:**

Yes

**Other Comments Or Suggestions:**

In definitions and lemmas, clearly explaining each variable would be helpful; additionally, providing accompanying causal diagrams would significantly enhance readability and understanding.

**Other Strengths And Weaknesses:**

**Strengths**:

-Compared to Twin Networks and k-plet Networks, AMWN reduces complexity by requiring fewer variables for representing counterfactual scenarios.

-Extends Pearl's do-calculus effectively to counterfactual reasoning.

**Weaknesses**:

Many definitions in the paper overly emphasize detailed explanations of variables. Although the paper compares AMWN to Twin Networks and SWIG, detailed experimental results quantifying performance improvements are not provided.

**Questions For Authors:**

Could you provide a simple example illustrating a scenario where your method succeeds but the approaches listed in Table 1 fail—for instance, demonstrating that the Twin network algorithm is not complete, while yours is?

**Relation To Broader Scientific Literature:**

The paper generalizes Pearl's celebrated do-calculus from interventional to counterfactual reasoning. The proposed AMWN improves upon existing frameworks, including Twin Networks, Single World Intervention Graphs, and Multi-Networks. Additionally, the three rules introduced for ctf-calculus are more general than Pearl’s do-calculus and the Potential Outcome Calculus (po-calculus).

**Theoretical Claims:**

I have checked most of the proofs. However, for Lemma 2.3, Definition 2.4, and Theorem 3.1, providing a causal diagram would greatly enhance clarity and ease of understanding.

---

> ### Author Rebuttal · Authors · 2025-04-01
>
> Thank you for reviewing our paper.
>
> To address your question about a scenario where our method succeeds but the other approaches mentioned in Table 1 fail, let us consider the question of whether the causal graph in Figure 4(b) implies that $(Y_{xw}, W_{x'} \perp X | {Z_x}')$. Figure 5(a) shows a 3-plet (triplet) network (a natural generalization of the twin network to 3 worlds) for this graph and question. The variables in the query involve three submodels: $\mathcal{M}, \mathcal{M_x}$, and $\mathcal{M_{xw}}$, all depicted in the network sharing explicit unobservable variables. It seems that $X$ is d-connected to $Y_{xw}$ given ${Z_{x}}'$, because there is an active path $X \gets Z \to U_z \to Z_{xw} \to Y_{xw}$. However, as discussed in the manuscript (around line 262) and due to exclusion restrictions, conditioning on $Z_{x'}$ is equivalent to conditioning on $Z$, meaning the corresponding separation statement holds. This implies that the Twin Network alone leads us to infer a wrong conclusion. Although this example involves three worlds (to take advantage of the figure in the paper), the same argument could be made with only two of them.
>
> For Single World Intervention Graphs (SWIGs), conditional independence among variables present in the SWIG can be read using d-separation, while the representation itself cannot capture cross-world restrictions on the counterfactual joint distribution. For instance, the separation of $X$ and $Y_x$ given Z cannot be judged using the SWIG for Figure 4(a) and intervention $X = x$ because Z does not appear in the resulting graph. These examples and some discussion on Shpitser & Pearl 2007 can be found in the supplemental material, section C.
>
> The k-plet Network is a concept we use in the paper to refer to the extension of the Twin Network method to k worlds (2-plet network equals Twin Network). In the paper, we imply that when combined with the exclusion operator, the k-plet network method is complete, but it can further be optimized, leading to the discussion of AMWNs. We added proper discussion and clarification about this point in the paper. Thank you!
> Also, as you pointed out, several definitions in the paper emphasize explanations about the notation and counterfactual variables. We will try to make those definitions more concise, maintaining the essence of the definition while moving additional explanations elsewhere.

---

> > ### Comment · Reviewer_GANy · 2025-04-02
> >
> > The authors have addressed my concerns. Hence, I will maintain my current score and lean toward accepting the paper.

---

### Decision · Program_Chairs · 2025-05-01

**Decision:**

Accept (spotlight poster)

**Comment:**

This paper introduces AMWNs, a novel graphical framework for counterfactual reasoning that is both sound and complete for d-separation-based counterfactual independence. Additionally, it proposes a counterfactual calculus that generalizes Pearl’s do-calculus to handle counterfactual queries, filling a longstanding gap in the causal inference literature.

All four reviewers gave positive assessments, highlighting the paper's theoretical depth, rigorous proofs, and its significance in advancing counterfactual inference. The proposed framework addresses limitations in Twin Networks and SWIGs by avoiding graph blow-up and supporting general counterfactual transformations. The rebuttal addressed the main concerns raised by reviewers.

While the paper does not include experiments—understandable for a theoretical contribution—some reviewers noted that clearer visualizations (e.g., Figure 3), improved explanations of certain concepts (e.g., nested ctf reduction), and minor editorial refinements could enhance accessibility. Additionally, a brief discussion comparing AMWN's d-separation with other graphical independence criteria could strengthen the broader contextual framing.

Overall, the paper presents a significant contribution to the theory of causal inference. I recommend acceptance.